# OpenReview forum: "RoboCook: Long-Horizon Elasto-Plastic Object Manipulation with Diverse Tools"
_robot-learning.org/CoRL/2023/Conference — CoRL 2023 Oral_

### Official Review · Reviewer_UkA9 · 2023-07-11

**Confidence:** 5
**Originality:** Good
**Technical Quality:** Good
**Clarity Of Presentation:** Good
**Impact:** 3

**Recommendation:**

Weak Accept: I recommend accepting the paper, but will not argue for my recommendation if the majority of other reviewers have a different opinion.

**Review:**

Strength:
* Challenging long-horizon and deformable object manipulation task;
* Extensive studies with comparison to baselines and a generalisation task.

Weakness:
* Limited method novelty;
* Unclear that how large the datasets are for a sufficient coverage of state space;
* Lacking insights to interpret the performance difference in Table 1.


**Quality Of The Limitations Section:**

Limitations are addressed clearly

**Questions For Rebuttal:**

* Figure 3: since the synthetic data-set is generated from randomly sampled action, it is unclear how large the dataset should be and how long the exploration should be to sufficiently cover states that are similar to target shapes. Does this random walk always start from the block-shaped state? Will this become a scalability issue if more tools or finer target shapes were considered?

* Line 133, "closed-loop control with visual feedback": it is not clear which control rate is the control operated on. Is it per step or per tool stage? Is it sufficiently efficient to run the mesh reconstruction and re-sampling in real-time to support feedback at this level?

**Robotics Focus:**

Sufficient demonstration on hardware

**Summary Of Paper:**

The paper presents a system that can choose appropriate tools to shape dough in making dumplings and letter-shaped cookies. The system builds upon a GNN learned from interaction data for each tools that model the dynamics, and a mesh registration and re-sampling process to map the visual observation to the particle representations. Instead of planning through the learned dynamics, the authors propose to learn the inverse dynamics given the goal shape by synthesising from the learned forward dynamics, thus significantly reduce the planning time for a closed-loop solution. An additional model is trained to infer the most probable tool to use facing the current observation and target shape. The results show that the learned dynamics is generalisable from dumpling making task to the cookies task and cookies shape with different materials.

**Summary Of Recommendation:**

The work presents an interesting system dealing with a task with considerable complexity. The proposed methods are not ground-bleakly novel while are reasonably composed and sufficiently tested in empirical studies. Some more in-depth discussion and analysis could make the framework stronger. However, its current form is interesting and thorough enough as a system paper.

---

### Official Review · Reviewer_bNTe · 2023-07-17

**Confidence:** 4
**Originality:** Good
**Technical Quality:** Excellent
**Clarity Of Presentation:** Very Good
**Impact:** 3

**Recommendation:**

Strong Accept: I recommend accepting the paper and will argue for my recommendation even if other reviewers hold a different opinion.

**Review:**

# Originality

## Problem Statement
The problem of robot learning for dough manipulation is relatively new, but with a reasonable amount of investigation in the last few years. It is an interesting and impactful area of research - however the authors are not the first to propose dumpling making or letter-shaping as tasks. Additionally, most/all of the individual subtasks proposed for dough manipulation (cutting, pushing, rolling) have been incorporated into robot systems before ([1], for instance).

[1] Planning with Spatial-Temporal Abstraction from Point Clouds for Deformable Object Manipulation, cited in the work.

## Methods:

The three components of the method are relatively well-understood techniques in the community:
Learning dynamics models of particles in deformables or liquids, using GNNs, is a common approach. However, the authors propose a very nice methodology for extracting clean+consistent point clouds suitable for training a dynamics model from real-world data.
Tool classification: the authors use a standard classification architecture to choose a tool - whether the training of this classifier used anything non-standard (self-supervised, or novel loss/dataset) was a bit unclear from the main paper.
Self-supervised, goal-conditioned action prediction: learning an inverse dynamics model to use as a goal-conditioned policy is a well-studied technique, as is learning parameterized skills for each tool. I haven’t specifically seen them used together, but it’s a natural design in a long-horizon, multi-skill setting where intermediate steps are well-understood.

I would classify this paper as a novel (and high-quality) arrangement of many standard learning+robotics techniques that enables very impressive dough manipulation tasks.

# Quality:
The quality of this submission is extremely high. The robotic system used is quite impressive:
1. the use of multiple cameras to extract a clean point cloud of the dough which is good-enough-quality for training a dynamics model is no small feat, even if some of the perception system is made easier using color segmentation.
2. The tool design and integration is quite extensive and well-thought-out, as are the interfaces between the robot+tool and tool+action space. And the tool swapping looked quite smooth. Overall an extremely impressive robotic setup.
3. Each submodule seems to have been implemented with care and thorough/thoughtful design.

Some small design issues I noticed:
1. Using the Earth-Mover’s distance as a part of the loss when you have occlusions between start/end frames seems like a recipe for difficult convergence / poorly-defined loss metric, since the start points are (presumably) evolved to their final locations via the GNN, even though the initial points may no longer be visible in the final observation.
2. The perception system relies quite heavily on knowledge of the environment, including color + geometry + full knowledge of tools.

Also, the videos are great.

# Clarity:
The submission is quite well-written, and well organized. There were a couple of unclear parts:
1. The authors don’t really discuss how the goal states are generated for each substep of the manipulation pipeline.
2. Using s=2 as a trajectory length doesn’t really feel like a long-horizon prediction: perhaps motivate this better in the main text?
3. How is the environment reset during random exploration?

# Significance:
I’d say this work is reasonably significant, because it provides a pretty compelling system which has impressive performance on a notoriously difficult manipulation task. While I don’t think that any of the learning components used in the system are substantially different from prior work (and many of the subcomponents have been used even in prior dough manipulation work), it plants an impressive flag - I don’t think I’ve seen a system which completely makes dumplings before with as few constraints on the dough (size, initial position, etc.) as in this work.

I think the design of the tools, their action spaces, and describing a full start-to-finish dumpling making procedure which uses them is in-and-of-itself valuable, because it provides an excellent starting point and point of reference for other researchers looking to improve a full working dumpling-making system. While it falls short of providing a benchmark set of subtasks (which I think would be a very cool contribution…), it would likely be easy for others to compare submodules with the performance of this system (and also ground the utility of others’ submodules).

# Relevance:
Definitely a relevant paper, because it contains both a real robot and real-world learning, and on a very challenging task. The CoRL audience will definitely be interested in the paper, although perhaps more for the system design than for any of the algorithms used/proposed.

# Limitations:
The paper lists a few limitations, which all boil down to the two following categories:
1. The policy is following a high-level script, and therefore cannot recover from failures (either system-wide or even back-tracking). For specific, noteworthy tasks like making this type of dumpling that might be fine, but modifying this script for other variations (different size dumplings, different crimping, different filling, etc.) would be highly manual.
2. Limitation to dough - other deformables with other constraints would require a very different perception + dynamics-modeling system.

There are two other substantial limitations not mentioned, in my opinion:
1. The perception system makes many simplifying assumptions which reduce it’s generality - knowing dough color, table shape, exact tool sizes, etc. Allow the authors to dodge difficult perception problems - and it’s not exactly obvious to me that the system would scale to more complex dough scenes (i.e. where there’s a lot of dough being worked with, cut up, etc.).
2. There’s a rather inherent limit on the expressivity of the sub-policies. Using primitives controlled at low-frequency with no real tactile/proprio or otherwise closed-loop feedback during the execution of the trajectory limits the complexity of tasks this system could be incorporated in. The only piece of the system which would really require this kind of feedback when executed by a human (crimping) is offloaded to a specialized tool (the metal crimping device).

# Strengths/Weaknesses
Strengths:
- Excellent technical system, thoughtfully executed
- Famously difficult domain with a challenging long-horizon task
- The system works!

Weaknesses:
- Not much novelty of sub-components
- The method evades some of the more-interesting problems in dough manipulation (i.e how to control dough w/ feedback at very-high frequency, maybe with additional sensory modalities) through careful design of tools and processes.
- Requires a human-defined script and subgoals.



**Quality Of The Limitations Section:**

Additional details required

**Questions For Rebuttal:**

Not so many questions, mostly just a request for more clarity on the following from above:

- The authors don’t really discuss how the goal states are generated for each substep of the manipulation pipeline.
- Using s=2 as a trajectory length doesn’t really feel like a long-horizon prediction: perhaps motivate this better in the main text?
- How is the environment reset during random exploration?


**Robotics Focus:**

Sufficient demonstration on hardware

**Summary Of Paper:**

In this work, the authors propose a real-world robotic system and learning framework for manipulating elasto-plastic objects (namely, dough) over long horizons and using tools. The paper introduces three components which they use as building blocks for the long-horizon policy. First, they propose to learn a forward dynamics model of dough/tool interaction from raw point-cloud input using a GNN. Second, they learn a model to select the appropriate tool for the current stage of the manipulation. Finally, they leverage the dynamics model to train a policy (may be viewed as an inverse dynamics model) which generates the action to move between two different dough states using a self-supervised learning procedure. The authors then design two long-horizon dough manipulation tasks (dumpling making and alphabet shaping), and convincingly demonstrate that their system is able to accomplish these difficult tasks in a real robot environment.


**Summary Of Recommendation:**

I really like this paper - it's a really strong example of a well-designed system which completes a very difficult task and incorporates learning methods in reasonable ways. The attention to detail in the system design, in particular, is something that many roboticists should aspire to. It's also the first time I've seen a full dumpling-making setup w/ learning which wasn't entirely scripted.

While the subcomponents don't have much technical novelty, their integration in a larger system which actually works warrants acceptance of this work to CoRL, if only to serve as inspiration for researchers in the robot learning community to consider any of the (many) intermediate steps of dumpling-making, which each highlight unique challenges which independently warrant investigation. Being able to see system-wide limitations is excellent (and doesn't happen nearly enough, as much research tends to be quite narrow, particularly in deformable manipulation).

---

### Official Review · Reviewer_WdqP · 2023-07-18

**Confidence:** 3
**Originality:** Excellent
**Technical Quality:** Excellent
**Clarity Of Presentation:** Excellent
**Impact:** 4

**Recommendation:**

Strong Accept: I recommend accepting the paper and will argue for my recommendation even if other reviewers hold a different opinion.

**Review:**

Strengths:
- This work tackles important, novel and challenging manipulation tasks.

- The paper is well-written and easy to follow. The figures did a good job illustrating the ideas.

- The system is reasonably designed with insightful discussion on the various choices.

- The experiments have been conducted extensively on real world hardware. The results are convincing.

- The additional results on generalizing to different materials without retraining is quite interesting.


Weakness and questions:
- If I were to be nitpicky, there is a lack of baseline for comparison in the task dumpling making. However, it is understandable that finding a suitable baseline on these complex tasks may not be easy.

- Another prior art on tool usage for long-horizon sequential manipulation [A] might worth discussion in the related work.

Overall, this is a solid work with multiple merits.

References:
    A) Liang, J., Wen, B., Bekris, K. and Boularias, A., 2022, May. Learning sensorimotor primitives of sequential manipulation tasks from visual demonstrations. In 2022 International Conference on Robotics and Automation (ICRA) (pp. 8591-8597). IEEE.

**Quality Of The Limitations Section:**

Limitations are addressed clearly

**Questions For Rebuttal:**

see above

**Robotics Focus:**

Sufficient demonstration on hardware

**Summary Of Paper:**

This work develops an intelligent robotic system, RoboCook, which perceives, models, and manipulates elasto-plastic objects with various tools. RoboCook uses point cloud scene representations, models tool-object interactions with Graph Neural Networks (GNNs), and combines tool classification with self-supervised policy learning to devise manipulation plans. It demonstrates that from just 20 minutes of real-world interaction data per tool, a general-purpose robot arm can learn complex long-horizon soft object manipulation tasks, such as making dumplings and alphabet letter cookies.

**Summary Of Recommendation:**

This is a solid work with multiple merits.

---

### Official Review · Reviewer_VnEP · 2023-07-24

**Confidence:** 4
**Originality:** Good
**Technical Quality:** Very Good
**Clarity Of Presentation:** Good
**Impact:** 3

**Recommendation:**

Weak Accept: I recommend accepting the paper, but will not argue for my recommendation if the majority of other reviewers have a different opinion.

**Review:**

Strengths:
- Excellent system demonstration; able to perform two difficult tasks (making dumpling and letter shaping).
- Method is well formulated, runs fast, and seemingly trained fully from randomized data.

Weaknesses:
- Relies upon provided subgoals to achieve long horizon planning.
- Human priors on actions space are difficult to scale to new tools/tasks.
- Paper is lacking important details w.r.t. the proposed method and the baseline.
- Fixed set of tools.

This work proposes a novel framework that builds on existing work modeling deformable object interactions with Graph Neural Networks (GNNs) (e.g., [18]) by adding a tool selection model that forwards to either a GNN based low-level policy or a hand-coded policy. This enables their approach to effectively perform many subtasks required to solve complex tasks such as making a dumpling.

The authors claim that RoboCook enables long-horizon tasks, however, their method seems to be highly reliant upon subgoal images that guide the robot though the task. As such, individual policies really perform short-term goals and no long-horizon reasoning seems to be taken place. The authors should clarify how RoboCook enables long-horizon reasoning.

The description of the proposed method is somewhat difficult to follow and is missing important details. The authors should be clearer in the introduction that their framework also uses hand-coded policies for certain tools (though the tool/policy is still autonomously selected). Further details on each component should be given, specifically, what exactly are the form of the inputs/outputs of each model component. For instance, are edge features considered in the GNN? Is the tool identity input to one network or is there a network for each tool? As a point of reference, [18] eqs. 2-5 help clarify the inputs/outputs of their GNN model; a similar style of network description could really help clarify the methodology here.

Further clarifications on the baseline methods are needed to clarify the significance of the results. In particular, more details on what data baselines 1 and 2 are implemented with (still multi-tool?) should be provided. The inputs/outputs of the RL baseline is also difficult to understand, even with the appendix description.

A limitation of this work is also the hand designed action spaces. Here, the class of task makes it so that actions can be easily specified, but generally speaking it can be difficult/undesirable to specify tool action primitives (e.g., consider multiple-objects in scene or full 6-dof control). This somewhat limits the impact of this work.

The proposed method is able to select between tools/policies effectively and all from self-supervised random interactions. Being able to learn autonomously from simple/small self-supervised data collection policies is a significant result for tool use. However, it does here assume guided subgoals to complete the longer task. Discussing how future work might loosen this dependency further could help solidify the significance of the work. Similarly, how could future work enable method to generalize to novel tools during runtime?

**Quality Of The Limitations Section:**

Limitations are addressed clearly

**Questions For Rebuttal:**

- What are the precise inputs/outputs for each component of RoboCook (e.g., GNN dynamics, tool predictor, and policy network)?
- On what data are the baseline models trained, in particular baselines 1 and 2? How long is the RL baseline trained and what are the precise inputs/outputs?
- How are subgoals collected here? Video seems to suggest that the method may be able to realize when it needs to backtrack to an earlier subgoal, but does not specify how this is/could be done.
- Is tool identity input to the network or is there an individual policy per tool?
- How are the bins for the action policy established?
- In what frame are particle locations localized in as inputs to the GNN? If global frame, how is translation invariance achieved?
- In Sec. 6.1.2, what is mean by "optimizing the sample quality at each time frame"?
- In Sec. 6.2.2, the "temporal abstraction" is unclear it unclear how this is related to lines 111-112 in terms of prediction horizon. Please provide clarification.

**Robotics Focus:**

Sufficient demonstration on hardware

**Summary Of Paper:**

This work proposes RoboCook, a framework for manipulating deformable material (e.g., dough) with a set of available tools. They propose a novel tool selection approach by self-supervising a classifier on random tool interaction results. Given a subgoal pointcloud, the classifier selects the tool and either a goal-conditioned tool-conditioned action policy or a hand-designed policy is executed to reach the subgoal. Policies are trained using random tool action rollouts. The experiments show that the method can be used (given subgoal images) to create a dumpling and shape dough into arbitrary shapes (e.g., letters) and outperforms several baselines on the latter task.

**Summary Of Recommendation:**

While I am not sure that their claim to "long-horizon planning" is justified, due to the prescribed subgoals, their full system, self-supervised to simultaneously train dynamics, policies, and the tool selection policy seems novel and effective for a difficult set of tasks and is nicely demonstrated with high performance compared to existing baselines. The paper needs some clarity on method details, but I believe presents a novel and significant result.

---

### Author Response · Authors · 2023-08-10
**A general response to common concerns**

We thank the reviewers for their detailed reviews. Reviewers agree that this work "shows an excellent system demonstration," "tackles important, novel, challenging manipulation tasks with considerable complexity," "presents a novel and high-quality arrangement of many standard learning+robotics techniques," "carries out extensive experiments on real-world hardware," and "presents convincing results and an extremely impressive robotic setup."

Here is a general response to several common concerns raised by reviewers:

> Not much novelty of sub-components in the system.

This is a system paper. We aim to use a novel combination of existing techniques to solve dumpling making, which is notably challenging in dough manipulation. This is the first time people have demonstrated that a general-purpose robot arm can accomplish long-horizon soft object manipulation tasks as complex as making dumplings. We have also shown the system's robustness by perturbing the robot during real-time execution in the supplementary video `02_perturbation.mp4`.

> How are subgoals collected in the dumpling-making task?

As mentioned in Section 6.1.1, during data collection, we execute a hand-coded dumpling-making pipeline ten times and add point clouds captured before and after using each tool in the pipeline to our dataset as expert demonstrations. The point clouds recorded after using each tool in one of these trajectories are selected as subgoals.

> How does RoboCook enable long-horizon reasoning?

RoboCook breaks down the long-horizon task into short-term subtasks and uses a tool predictor for high-level task planning. The closed-loop tool predictor effectively enables RoboCook to reason in long-horizon tasks. When the dough state changes, our tool predictor takes in the visual feedback and predicts the best tool at the current stage to achieve the final target, so it does not necessarily follow a fixed sequence of skills and naturally knows when to backtrack and when to skip.

> Hand-designed action spaces seem undesirable in general manipulation settings.

We agree that hand-designed action spaces are not desirable in general manipulation settings. Our insight is that for most tasks, especially tasks where the robot uses a tool, there is much redundant and undesired space in the full 6-DoF action space. In most cases, humans also adhere to certain rules and follow a constrained action space when using a specific tool. A future direction is to design a planner to automatically prune the action space conditioned on the tool and the task.

---

### Decision · Program_Chairs · 2023-08-30

**Decision:**

Accept (Oral)

**Comment:**

This work proposes a method for manipulating deformable objects with a set of tools. The authors propose a classification-based tool selection method, which, given a target point cloud, selects the tool and a policy. The experiments demonstrate that the method outperforms baselines for dough-shaping tasks.

Strengths:
- A strong set of experiments shows the method works for two difficult tasks
- Especially impressive is the extensive use of real-world hardware
- It is impressive that the method works while only being trained on random data
- The paper is well-written.

Weaknesses:
- Subgoals are necessary to achieve non-trivial tasks
- Some domain knowledge regarding action spaces is needed
- Unclear if this method will generalize to new tools

Please see the reviewers' comments for more information.